# Identification of the Interacting Domains Between Tissue Factor and β1-Integrin and the Signalling Properties of the Two Fibronectin-like Domains of Tissue Factor

**DOI:** 10.3390/cancers17040644

**Published:** 2025-02-14

**Authors:** Sophie J. Featherby, Eamon C. Faulkner, Anthony Maraveyas, Camille Ettelaie

**Affiliations:** 1Centre for Biomedicine, Hull-York Medical School, University of Hull, Cottingham Road, Hull HU6 7RX, UK; e.faulkner@hull.ac.uk (E.C.F.); c.ettelaie@hull.ac.uk (C.E.); 2Clinical Sciences, Hull-York Medical School, University of Hull, Cottingham Road, Hull HU6 7RX, UK; anthony.maraveyas@nhs.net

**Keywords:** tissue factor, β1-integrin, fibronectin-like domains, EGF4 domain, βTD domain, cell signalling, cell proliferation

## Abstract

The coagulation protein tissue factor (TF) is upregulated in a number of cancer types and can activate signalling pathways, which promote cancer progression. One of the mechanisms by which TF induces signalling in cancer cells is by interacting with the cell adhesion molecule β1-integrin. However, the domains within the proteins responsible for complex formation are not known. Examination of the nature of the interaction between TF and β1-integrin suggested that the upper fibronectin-like domain of TF interacted with the EGF4 domain of β1-integrin and induced signalling that resulted in cell proliferation. In addition, the lower fibronectin-like domain of TF robustly interacted with the βTD domain of β1-integrin reinforcing complex formation. These findings suggest a mechanism by which TF aids cancer cells in sustaining adhesion-free survival and may provide an explanation of why tumours that express high levels of TF exhibit aggressive and malignant phenotypes.

## 1. Introduction

Tissue Factor (TF) is a 47 kDa transmembrane glycoprotein receptor [1] that has a 219 amino acid extracellular domain consisting of two fibronectin-like domains, a transmembrane domain and a short 21 amino acid cytoplasmic domain [2,3]. TF is best known as the primary cellular initiator of blood coagulation [4,5]. However, TF is also capable of initiating coagulation factor-dependent and -independent cellular signalling [6]. TF is upregulated in many cancer types, and cellular signalling initiated by TF is known to contribute to cancer development and progression [7]. One of the mechanisms by which TF initiates cell signalling is by interacting with β1-integrin [8].

Integrins are cell surface receptors that bind to extracellular matrix (ECM) proteins, functioning as adhesion molecules involved in cell–ECM and cell–cell interactions [9]. Integrin complexes are formed from non-covalently associated heterodimers of α and β subunits, with β1-integrin being the predominantly expressed β subunit in both normal and cancerous cells [10]. TF was previously shown to interact with β1-integrin using co-localisation and co-immunoprecipitation experiments [11,12,13,14]. However, the domains within both TF and β1-integrin proteins responsible for complex formation are not known. Functionally, the interaction between TF and β1-integrin has been shown to induce cell migration, proliferation and angiogenesis [15,16,17,18,19]. For example, the overexpression of both full-length TF and alternatively spliced (as)TF in MCF-7 breast cancer cells resulted in increased cell proliferation, which was reduced following the knockdown of β1-integrin expression [17]. Since asTF contains only the extracellular domains of TF, it is likely that the extracellular region of the protein is responsible for interacting with β1-integrin [17,18,20]. When cells are treated with the monoclonal anti-TF antibody 10H10, the interaction between TF and β1-integrin is disrupted, preventing their co-immunoprecipitation [14]. Crystal structure analysis of the 10H10 Fab in complex with TF shows that the antibody binds at the interface between the two fibronectin-like domains of the TF extracellular domain [21], further implicating these regions in the binding of β1-integrin. Additionally, the treatment of primary human endothelial cells with recombinant TF resulted in increases in Extracellular Signal-Regulated Kinases (ERK)1/2 phosphorylation and accelerated cell proliferation [15,16]. The increase in proliferation was prevented by pre-incubation of the recombinant TF with a peptide corresponding to the membrane-proximal 4th Epidermal Growth Factor-like domain repeat (EGF4) and β-tail domain (βTD) of β1-integrin [15,16] or a blocking antibody targeting the same region [17]. Therefore, it is proposed that the EGF4 and βTD domains within the β1-integrin protein act as the binding site for TF.

Although the functional outcomes of interactions between TF and β1-integrin have been reported, the mechanism by which the proteins induce cellular signalling has not been investigated. It is known that the binding of integrins to ECM proteins induces conformational changes in the integrin proteins [22]. Such conformation changes in β1-integrin activate signalling cascades, which have been implicated in the regulation of the survival, proliferation, differentiation, migration, and apoptosis of cells [9,23,24]. We hypothesised that the interaction between TF and β1-integrin may promote cell proliferation by inducing a conformational change in β1-integrin [11,17]. This study aimed to identify the domains within both TF and β1-integrin proteins responsible for complex formation. In addition, the mechanism by which the interaction between the two proteins influenced cellular signalling and its effect on cell proliferation was investigated.

## 2. Materials and Methods

### 2.1. Preparation of Plasmid Constructs for the Expression of TF and β1-Integrin Domains

Prior to experiments, the cDNA corresponding to the complete extracellular domain of TF (residues 1–219; TED), the upper fibronectin-like domain of TF extracellular domain (residues 1–110; UED) and the lower fibronectin-like domain (residues 106–219; LED) (Figure 1A) were cloned into the FLAG-HA-pcDNA3.1 expression plasmid (Addgene, LGC Standards, Teddington, UK). The PCR-based cloning procedure employed the following sets of primers: TED forward primer 5′-tcg cct cta gag ccg gat cct cag gca ct-3′ and reverse primer 5′-atg tag cgg ccg cct att ctc tga att ccc ctt t-3′, UED forward primer 5′-tcg cct cta gag ccg gat cct cag gca ct-3′ and reverse primer 5′-ctc tgg cgg ccg cct atg gct gtc cga ggt ttg t-3′ and LED forward primer 5′-tcg cct cta gag ccg gat cct cag gca ct-3′ and reverse primer 5′-ctc tgg cgg ccg cct atg gct gtc cga ggt ttg t-3′ (Eurofins, Wolverhampton, UK). The pCMV6-AC turbo-GFP-tagged plasmid containing human TF cDNA (F3 gene NM_001993) (Origene/Cambridge bioscience, Cambridge, UK) was used as the template DNA for PCR reactions. In addition, the cDNA corresponding to the EGF4 domain (residues 572–610) and βTD (residues 611–728) (Figure 1B) of β1-integrin were amplified using forward primer 5′-gga tct aga ggt gtt tgc aag tgt cgt-3′ and reverse primer 5′-aca atg gat cct tag tct gga cca gtg gga c-3′ (Eurofins) from the pCMV6-XL5 plasmid containing human β1-integrin cDNA (ITGB1 gene NM_002211) (Origene). All amplified sequences were digested with the XbaI restriction enzyme together with either NotI or BamHI (Promega, Southampton, UK) and were then ligated into the FLAG-HA-pcDNA3.1 expression plasmid using the NEB instant sticky ends ligase master mix (New England Biolabs, Hitchin, UK). Successful ligation of the plasmid constructs was confirmed by sequencing (Eurofins).

### 2.2. Cell Culture and Transfection of Plasmid Constructs

The MDA-MB-231 breast cancer cell line was obtained from CLS (Köln, Germany) and cultured in the DMEM medium (Lonza, Cambridge, UK) containing 10% (*v*/*v*) FCS (Gibco/Fisher Scientific, Paisley, UK). Primary human dermal blood endothelial cells (HDBEC) were obtained from PromoCell (Heidelberg, Germany) and cultured in MV media containing 5% (*v*/*v*) FCS and growth supplements (PromoCell). The cells were transfected with the FLAG-HA-pcDNA3.1 expression plasmid containing EGF4-βTD, EGF4, TED, LED or UED DNA, or with an empty control plasmid, using TransIT 2020 transfection reagent (Mirus Bio/Cambridge bioscience, Cambridge, UK). Cells transfected with the empty plasmid expressed a FLAG-HA control peptide. The cells were incubated for 48 h to permit the expression of the protein constructs and analysed as outlined below. In the stated experiments, cells were pre-incubated with a blocking anti-β1-integrin antibody (10 μg/mL; AIIB2, Merck Millipore, Burlington, MA, USA) for 16 h prior to harvesting.

The successful transfection of cells was confirmed by Western blot analysis using a rabbit anti-HA-tag antibody (GeneTex, Isleworth, UK) and goat anti-rabbit alkaline phosphatase-conjugated antibody (Santa Cruz Biotechnology, Heidelberg, Germany) using the procedure described below. The excretion of the fragments from the cells into the cell culture media was also assessed. Cell culture media was collected 48 h after transfection and centrifuged at 8000× *g* for 5 min to remove cell debris. The media was then added to either a Microsep centrifugal filter (1 kDa cutoff; Pall Filtron/Flowgen, Lichfield, Staffordshire, UK) or a Centricon concentrator (3 kDa cutoff; Amicon, Beverly, MA, USA) as appropriate and centrifuged at 3000× *g* for up to 5 h to concentrate proteins. The concentrated media was then analysed by Western blot using a rabbit anti-HA tag antibody (GeneTex) and goat anti-rabbit alkaline phosphatase-conjugated antibody (Santa Cruz) using the procedure described below. Finally, the presence of the protein constructs on the cell surface was confirmed by immunofluorescence microscopy analysis of non-permeabilised cells. Transfected cells were incubated for 48 h, and then the cells were fixed with 4% paraformaldehyde for 10 min. The cells were probed with a rabbit anti-HA-tag antibody (C29F4; Cell Signalling Technology, Leiden, The Netherlands) and then developed with NL493-conjugated anti-rabbit IgG antibody (R & D Systems Europe Ltd., Abingdon, UK). Images were acquired using a Zeiss Axio Vert.A1 inverted fluorescence microscope (Carl Zeiss Ltd., Welwyn Garden City, UK) at ×40 magnification.

### 2.3. Analysis of Protein Interaction by Proximity Ligation Assay (PLA)

The interactions between expressed peptides and cellular proteins were assessed using Duolink PLA reagents (Sigma Chemical Company Ltd., Poole, UK). MDA-MB-231 cells (10^4^) or HDBEC (5 × 10^4^) were plated in 29 mm culture dishes with a 10 mm glass-bottomed micro-well (InVitro Scientific/Cellvis, Sunnyvale, CA, USA). The cells were incubated with mouse anti-HA-tag antibody (1:50 *v*/*v*; C29F4; Cell Signalling) paired with either rabbit polyclonal anti-TF antibody (1:50 *v*/*v*; ab104513; Abcam, Cambridge, UK) or rabbit polyclonal anti-β1-integrin antibody (1:50 *v*/*v*; GTX112971; GeneTex) overnight at 4 °C. For comparison, positive controls were prepared using a mouse anti-TF antibody (1:50 *v*/*v*; HTF1; eBioscience/Thermo Scientific, Warrington, UK) and a rabbit anti-TF (1:50 *v*/*v*; FL295; Santa Cruz) antibody, as well as a mouse anti-human TF antibody (1:50 *v*/*v*; HTF1) and a rabbit polyclonal anti-β1-integrin antibody. Negative controls were prepared using the mouse anti-HA-tag antibody (1:50 *v*/*v*; C29F4; Cell Signalling) paired with a rabbit IgG isotype control antibody (1:50 *v*/*v*; Santa Cruz). The cells were washed three times with PBS and assessed using the PLA reagents (Sigma), as previously described [25,26]. Finally, the cells were stained with DAPI (2 μg/mL; Sigma) and Phalloidin-iFluor 488 (2 µg/mL; Abcam). Images were acquired using a Zeiss Axio Vert.A1 inverted fluorescence microscope (Carl Zeiss Ltd.) at ×40 magnification. The number of red-fluorescent events and blue-fluorescent nuclei was determined using ImageJ software version 1.53t (U.S. National Institutes of Health, Bethesda, MD, USA) [26,27].

### 2.4. Analysis of Protein Interaction by Co-Immunoprecipitation

The interaction between the expressed peptides and cellular protein was assessed by co-immunoprecipitation as previously described [26]. Briefly, cells were lysed in PhosphoSafe buffer (150 µL; Merck-Millipore, Watford, UK) containing a 1% (*v*/*v*) protease inhibitor cocktail (Sigma). The lysates were incubated with the anti-FLAG-tag antibody (4 µg; C29F4; Cell Signalling) at 4 °C overnight to capture the recombinant protein constructs. Pure proteome protein A-magnetic beads (10 µL; Merck-Millipore) were added to all samples and incubated at 4 °C for 90 min. A sample of lysate was also incubated with the protein A-beads alone. The magnetic beads were washed five times with PBST (PBS containing 0.1% *v*/*v* Tween-20), and bound proteins were then eluted by heating in Laemmli buffer (70 µL) (Sigma) at 95 °C for 5 min. Samples were then separated by SDS-PAGE, transferred to nitrocellulose membranes and probed with the anti-TF antibody (1:3000 *v*/*v*; HTF-1; eBiosciences/Thermo Scientific) or anti-β1-integrin antibody (1:3000 *v*/*v*; GTX112971; GeneTex/Insight Biotechnology Ltd., Wembley, UK) as appropriate. Bands were then visualised using the Western Blue stabilised alkaline phosphatase-substrate (Promega Corp., Southampton, UK), recorded and analysed using ImageJ.

### 2.5. Assessment of the Conformation of β1-Integrin Using Antibody Binding Assay

MDA-MB-231 cells (10^4^) and HDBEC (5 × 10^4^) were plated in 96-well plates and transfected to express TED, LED, UED or control peptides as above. The cells were fixed with 4% formaldehyde for 20 min at room temperature. The cells were then washed three times with PBST and quenched (0.1% *w*/*v* sodium azide, 1% *v*/*v* hydrogen peroxide diluted in PBST) for 10 min at room temperature. The wells were washed three times with PBST and then incubated with the rabbit polyclonal anti-β1-integrin antibody specific for the EGF4-βTD domains (1:100 *v*/*v* in PBST; PA1-37318; ThermoFisher, Loughborough, UK) for 90 min at room temperature. Additionally, test samples were incubated with monoclonal 9EG7 (1:100 *v*/*v*; BD BioSciences, Wokingham, Berkshire, UK) or AIIB2 (1:100 *v*/*v*; Merck Millipore) antibodies, which selectively recognise the active/open conformation [28] or the folded/inactive conformation [29,30] of β1-integrin, respectively. The wells were then washed three times with PBST and then incubated with HPR-conjugated antibodies against rabbit or rat IgG (diluted 1:100 *v*/*v* in PBST) for 1 h at room temperature. The cells were washed three more times with PBST and then incubated with TMB-one solution HRP substrate for 10 min at room temperature in the dark. The reactions were stopped using 2M sulphuric acid and the absorptions of the reaction solutions were measured at 450 nm using a PolarStar Optima plate reader (BMG Labtech Ltd., Aylesbury, UK). All values were normalised to the number of cells in each well, as determined by crystal violet assay measurements.

### 2.6. Measurement of ERK1/2 Phosphorylation by SDS-PAGE and Western Blot Analysis

Cells were lysed in Laemmli buffer (Sigma) and separated on denaturing 12% (*w*/*v*) polyacrylamide gels. Proteins were transferred onto nitrocellulose membranes and blocked with TBST (10 mM Tris–HCl pH 7.4, 150 mM NaCl, 0.05% Tween-20) for 1 h. On each occasion, the Western blot membrane was cut into two sections, and the higher molecular weight section was probed for ERK1/2 or pERK1/2 while the lower section was probed for GADPH. Western blot analysis of ERK1/2 phosphorylation in the samples was carried out using an anti-phosphoT202/185-phosphoY204/187-ERK1/2 antibody (Cell Signalling), and total ERK1/2 was detected using an anti-ERK1/2 antibody (Cell Signalling) diluted 1:3000 (*v*/*v*) in TBST. GAPDH was detected using a goat anti-GAPDH polyclonal antibody (V-18; Santa Cruz) diluted 1:5000 (*v*/*v*) in TBST. The membranes were incubated overnight at 4 °C and then probed with a goat anti-rabbit alkaline phosphatase-conjugated antibody or a donkey anti-goat alkaline phosphatase-conjugated antibody (Santa Cruz), diluted 1:5000 (*v*/*v*) in TBST. Bands were then visualised using the Western Blue stabilised alkaline phosphatase-substrate (Promega), recorded and analysed using ImageJ.

### 2.7. Preparation of cDNA and Quantification of Cyclin D1 Expression by qPCR

Cells were lysed, and the mRNA was extracted and converted to cDNA using the cell-2-cDNA kit (Ambion, Chipping Norton, UK) according to the manufacturer’s instructions. The relative quantity of cyclin D1 mRNA was quantified by qPCR using the SYBR Select Master Mix (Applied Biosystems, Warrington, Cheshire, UK) using the primers 5′-CCG TCC ATG CGG AAG ATC-3′ (forward) and 5′-ATG GCC AGC GGG AAG AC-3′ (reverse) [31]. The data were normalised against the housekeeping gene β-actin, which was amplified using a QuantiTect primer set (sequence not disclosed; Qiagen, Manchester, UK). The reactions were performed at an annealing temperature of 60 °C using an iCycler thermal cycler (Bio-Rad, Hemel Hempstead, UK) for 40 cycles. All changes in cyclin D1 mRNA expression were calculated using the 2^−ΔΔCq^ method in relation to the cells transfected with the control plasmid [32].

### 2.8. Determination of Cell Numbers by Crystal Violet Assay

Cell numbers were assessed by crystal violet staining using a kit obtained from Active Motif (La Hulpe, Belgium) as previously described [33,34]. Cells were fixed with 3% (*v*/*v*) glutaraldehyde for 30 min and stained with crystal violet solution (0.02% *w*/*v*) for a further 30 min. Cells were washed 3 times with PBS and the stain was then eluted from the cells via incubation in 1% (*w*/*v*) sodium dodecyl sulphate solution for 10 min. Absorptions were measured at 590 nm using the PolarStar Optima plate reader (BMG labtech Ltd.), and cell numbers were interpreted from standard curves constructed separately for each cell type.

### 2.9. Statistical Analysis

The presented data include the calculated mean values ± the calculated standard error of the mean. Statistical analysis was carried out using the GraphPad Prism version 9.0 (GraphPad Software, Boston, MA, USA). Significance was determined using one-way ANOVA (analysis of variance) and Tukey’s honesty significance test or, where appropriate, by a paired *t*-test, and *p* values of equal or less than 0.05 were deemed to be significant.

## 3. Results

### 3.1. Identification of the Interacting Domains Between TF and β1-Integrin by PLA and Co-Immunoprecipitation

The first aim of this study was to determine the interacting regions within the extracellular domain of TF and the membrane-proximal domains of β1-integrin. To this end, FLAG-tagged constructs encoding either the complete extracellular domain of TF (residues 1–219; TED), the upper fibronectin-like domain (residues 1–110; UED) or the lower fibronectin-like domain (residues 106–219; LED) were prepared (Figure 1A). The constructs were expressed in MDA-MB-231 cells, which express high levels of TF, and endothelial cells (HDBEC), which otherwise lack TF (Appendix A). The successful expression of the protein constructs by the cells was confirmed by Western blot analysis (Appendix A). Additionally, the release of the recombinant proteins was verified by detecting the constructs in the cell culture media (Appendix A). The observed molecular weights of the protein constructs in both cell lysates and culture media suggested that the recombinant proteins had undergone post-translational modification. Additionally, immunofluorescent staining of the TED construct was carried out (Appendix A). The use of intact cells, which had not been treated with the permeabilization agent, confirmed the construct was present on the cell surface. Finally, the endogenous expression of β1-integrin in both MDA-MB-231 cells and HDBEC was confirmed by Western blotting (Appendix A). Subsequently, any association between the constructs and cell-surface β1-integrin was assessed by the proximity ligation assay (PLA). The LED, UED and TED constructs were all associated with cellular β1-integrin on HDBEC (Figure 2). However, on MDA-MB-231 cells, the association of the UED construct with β1-integrin was marginally lower than the LED and TED constructs (Figure 3A,B). The ability of these constructs to bind to β1-integrin was also assessed by co-immunoprecipitation. The immunoprecipitation of the MDA-MB-231 cell lysate indicated that only the binding of the LED construct was sufficiently robust to co-purify the β1-integrin protein (Figure 3C,D; Appendix A).

To examine the ability of the membrane-proximal domains of β1-integrin to interact with TF, constructs encoding the combined EGF4-βTD domains (residues 572–728) and the EGF4 domain (residues 572–610) were expressed in MDA-MB-231 cells (Figure 1B). However, attempts to express a βTD construct were not successful. PLA examination of the cells indicated a significant level of association between the EGF4-βTD construct, but not the EGF4 domain alone, and cell-surface TF (Figure 4A,B). This was confirmed by co-immunoprecipitation of the TF from MDA-MB-231 cell lysate along with the FLAG-tagged EGF4-βTD construct (Figure 4C,D).

Finally, the ability of the expressed TED, LED and UED constructs to outcompete the binding of an antibody specific to the EGF4-βTD domains of β1-integrin (residues 579–799) was assessed. The expression of the LED construct in HDBEC significantly reduced the availability of the 579–799 aa region to the anti-β1-integrin antibody, but the influence of TED and UED was marginal (Figure 5). Similarly, the expression of all three peptides marginally reduced the availability of the site within β1-integrin in MDA-MB-231 cells. Taken altogether, these findings suggest that the extracellular domains of TF, along with the EGF4-βTD domains of β1-integrin, are the interacting domains between the two proteins. Moreover, the data indicate that LED may interact more strongly with β1-integrin than the UED, highlighting a distinct property of this lower domain.

### 3.2. Characterisation of the Influence of Expressed TF Domains on the Conformation of β1-Integrin

Next, the influence of the interactions between TF and β1-integrin on the structural conformation of the latter protein was examined. Cells expressing the TED, UED and LED constructs were probed with either 9EG7 antibody, which selectively binds β1-integrin in the active/open conformation [28], or with AIIB2 antibody, which binds to the inactive/closed conformation of the protein [29,30]. Subsequent analysis of the bound antibodies using the HRP-conjugated secondary antibody indicated that the expression of the UED construct significantly reduced the proportion of β1-integrin in the active/open conformation on the surface of MDA-MB-231 cells (Figure 6A), but did not significantly alter the amount in the inactive/close conformation (Figure 6B). Furthermore, both UED and LED constructs reduced the proportion of β1-integrin in the active/open conformation in HDBEC (Figure 6C), while the expression of TED preserved β1-integrin in the inactive/closed conformation (Figure 6D). These data show that the interaction of the extracellular domains of TF with β1-integrin can induce a conformational change in the latter protein. This finding may provide a potential mechanism by which the interaction of TF and β1-integrin can alter cellular signals.

### 3.3. Analysis of the Outcome of the Expression of the Construct Peptides on Cellular Proliferation

In order to examine the influence of the expressed protein constructs on proliferative signalling and cell proliferation, the phosphorylation of ERK1/2 protein, the expression of cyclin D1 mRNA and changes in cell numbers were quantified. Since the cleavage of protease-activated receptor 2 (PAR2) by the TF-factor VIIa (fVIIa) complex can lead to the activation of all three major MAPK pathways [35], it was necessary to rule out the possibility that the TF extracellular domain peptides were influencing cellular signalling via proteolytic mechanisms. Assessment of the peptides expressed by the cells indicated that neither TED, UED nor LED were capable of acting as a co-factor for fVIIa to sustain factor Xa (fXa) generation (Appendix A).

The expression of either UED or EGF4-βTD constructs in MDA-MB-231 cells resulted in reduced ERK1/2 phosphorylation (Figure 7A,B) and cyclin D1 expression (Figure 7C). The expression of UED also decreased the rate of cell proliferation (Figure 7D and summarised in Table 1). This reduction in proliferative signalling may result from the UED and EGF4-βTD constructs competing with the binding of the exogenously expressed TF and β1-integrin. Interestingly, the reductions in ERK1/2 phosphorylation and cyclin D1 expression caused by UED expression were not reduced further by prior inhibition of β1-integrin with the inhibitory AIIB2 antibody (Figure 8A–C). The finding that dual treatment with both UED construct and AIIB2 did not have a cumulative influence on proliferative signalling suggests that the UED construct and AIIB2 are inhibiting proliferative signalling via the same pathway. In contrast, the expression of the UED and LED constructs resulted in a small induction in the phosphorylation of ERK1/2 (Figure 9A) in HDBEC. However, the magnitude of these signals was not sufficient to translate into increases in either cyclin D1 expression or cell numbers (Figure 9B,C and summarised in Table 1). The expression of the EGF4-βTD construct did not influence the signalling in these cells since HDBEC do not express TF under physiological conditions. Collectively, these data indicate that the UED and EGF4-βTD constructs inhibit proliferative signalling in MDA-MB-231 cells, potentially by disrupting the constitutive TF–β1-integrin complexes presence in these cells. In contrast, in TF-deficient HDBEC, the expression of these constructs does not significantly alter cell signalling or proliferation, suggesting a context-dependent effect of TF–β1-integrin interactions.

## 4. Discussion

The coagulation protein TF is upregulated in a number of cancer types [7], and the expression of the protein is associated with poor prognosis [36,37]. One of the mechanisms by which TF contributes to cancer progression is through the interaction with the cell adhesion molecule β1-integrin [8]. The interaction between TF and β1-integrin influences multiple cancer processes, including promoting the proliferation of breast cancer cells [17]. Given that TF and β1-integrin have been shown to co-immunoprecipitate [11,13,14,19,38], their interaction is likely mediated through complex formation rather than via indirect mechanisms. However, the domains within the two proteins, which are involved in complex formation, have not been examined. In this study, it was demonstrated that both fibronectin-like domains within the extracellular region of TF were implicated in the interaction with β1-integrin. Furthermore, the EGF4 and βTD domains in β1-integrin are likely to participate in the interaction. The identification of these domains permits a proposed model for the complex in which the upper fibronectin-like domain of TF interacts with the EGF4 domain of β1-integrin and the lower fibronectin-like domain with the βTD domain (Figure 10A).

The involvement of the βTD domain of β1-integrin in complex formation proposes a mechanism through which TF influences integrin signalling [17]. The βTD in all β-integrin subunits contains a structural loop which comes into contact with residues within the head group of the β-integrin (Figure 10B). The interaction with the structural loop may be able to retain the integrin subunit in the inactive configuration [39,40]. Kocaturk et al. (2013) previously proposed that the interaction of asTF with the βTD region of the integrin disrupts/weakens the interaction between the βTD and the integrin head group allowing β-integrin to adopt an active conformation [17]. In our study, the expression of the EGF4-βTD peptide in MDA-MB-231 cells resulted in a reduction in pro-proliferative ERK phosphorylation and cyclin D1 expression. This observation suggests that the expressed EGF4-βTD peptide competes with full-length endogenous-β1-integrin for binding to cell-surface TF. Displacing cellular β1-integrin from TF may then allow the structural loop to re-engage with the integrin head group, promoting β1-integrin to return to an inactive conformation, thereby reducing proliferative signalling. These observations agree with previous reports showing that an EGF4-βTD peptide (residues 579–799) was capable of reducing the rate of proliferation in MDA-MB-231mfp cells and MCF-7 cells [17]. In addition, treatment of endothelial cells with the EGF4-βTD peptide (residues 579–799) reduced TF-mediated ERK phosphorylation and cell proliferation [15]. Collectively, these findings support the hypothesis that the interaction between TF and the βTD domain of β1-integrin can induce integrin signalling, offering a potential mechanism for the modulation of cell proliferation by TF.

Subsequently, the domains within TF that participate in TF–β1-integrin signalling were investigated. The expression of UED peptide in MDA-MB-231 cells suppressed proliferative signalling. These data suggest that, as with the EGF4-βTD fragment, the UED peptide may be capable of competing with TF for β1-integrin binding reducing the rate of cell proliferation. Conversely, the LED and TED peptides did not affect either ERK signalling or cyclin D expression. Therefore, while these peptides were shown to bind β1-integrin on the cell surface, they may not be able to displace endogenous TF and influence β1-integrin configuration. Alternatively, the LED and TED peptides may themselves be capable of sustaining a proliferative signal through binding to β1-integrin. In the latter scenario, the lower fibronectin-like domain, which is present in both LED and TED peptides, may interact with the structural loop in the βTD, allowing β1-integrin to attain an active conformation and promote proliferation. Therefore, although the extracellular domain of TF has been confirmed to contribute to β1-integrin binding, further studies are needed to identify the specific regions and residues within the upper and lower fibronectin-like domains that are involved in the interaction.

In agreement with the hypothesis that TF binding allows β1-integrin to adopt an active conformation, TF has previously been shown to co-immunoprecipitate with the active form of β1-integrin but not the inactive form [19]. In contrast, there is also evidence that the binding of TF to β1-integrin results in the inactivation of the latter protein. For example, in our study, the expression of the TED construct in HDBEC increased the proportion of β1-integrin in the inactive conformation, whilst the expression of the UED construct in MDA-MB-231 cells and HDBEC reduced the amount of β1-integrin detected in the active configuration. In another study, the treatment of cells with a monoclonal antibody 10H10, which disrupts the interaction of TF and β1-integrin, increased integrin activation [41]. Although, it should be noted that the conformational state of β1-integrin was not determined in this study. Furthermore, in the same study, evidence suggests that the binding of 10H10 antibody does not disrupt interactions of TF with all integrins but shifts TF complexing from β1-integrin to β4-integrin [41]. Additionally, co-immunoprecipitation experiments have shown that TF also interacts with αvβ3 [11]. Both β3 and β4-integrin are reported to contribute to cancer cell proliferation and survival [42,43,44,45]. Therefore, further research is needed to explore the interaction of TF with these different integrins and how these interactions influence cell behaviour in both physiological and cancerous contexts. Finally, it has also been reported that the interaction of the TF/fVIIa complex with β1-integrin can diminish integrin signalling by promoting the internalisation of the β1-integrin protein [19,38]. Therefore, regulation of the conformation of β1-integrin may not be the only mechanism by which TF–β1-integrin interactions influence integrin signalling.

The ability of TF to promote the active conformation of β1-integrin results in the initiation of integrin signalling that may be via a pathway that is independent of the ligation with ECM. Under physiological conditions, cells that lack integrin-dependent adhesion to the ECM undergo a form of apoptosis known as anoikis [46]. Therefore, cancer cells that upregulate both TF and β1-integrin would evade anoikis and may attain enhanced survival, independently of ECM adhesion. This adhesion-independent survival is crucial for cancer metastasis. Notably, the TF/fVIIa complex has been shown to inhibit anoikis in baby hamster kidney cells [47]. However, it is important to note that fXa, the downstream protease of the TF/fVIIa complex, is also capable of attenuating anoikis [47]. In contrast, disruption of the interaction between TF and β1-integrin with the 10H10 monoclonal antibody did not increase anoikis in MDA-MD-231mfp cells grown in vitro [14]. Therefore, TF may be capable of suppressing anoikis via PAR activation, as well as through β1-integrin mediated mechanisms.

This study explored the protein–protein interaction mechanisms that contribute to proliferative signalling arising from this interaction. However, it must be emphasised that the complex nature of the interaction of TF with β1-integrin influences several cellular properties and needs further investigation. Our study has attributed putative functions for each of the two fibronectin-like domains within TF, as mediators in proliferative signalling. Since the recombinant domains were able to reduce the proliferative signal in MDA-MB-231 cells, it may be envisaged that the expression of the recombinant proteins may have a similar impact in vivo. However, it is more likely that identification of the amino acids involved in the interaction between TF and β1-integrin would permit a more targeted inhibition of proliferative property associated with TF. For example, targeted antibodies or small disrupting molecules may be used to reduce cell proliferation in diseases such as cancer or, alternatively, to enhance growth to promote tissue repair

In conclusion, this study indicates a model of interaction between TF and β1-integrin in which the upper fibronectin-like domain of TF interacts with the EGF4 domain of β1-integrin and the lower fibronectin-like domain with the βTD domain. Furthermore, this interaction may alter the conformation of β1-integrin to one that is capable of inducing cellular signalling. Importantly, the induction of signalling via this pathway appears to be independent of the ligation of β1-integrin to the ECM. This, in turn, may explain how the expression of TF confers the ability to sustain adhesion-free cell survival in cancer cells, despite the loss of cell adherence [47]. This mechanism would facilitate cancer progression by allowing increased cell mobility and metastases [24,47] and explains why tumours which express high levels of TF exhibit aggressive and malignant phenotypes [36,37].

## Figures and Tables

**Figure 1 cancers-17-00644-f001:**
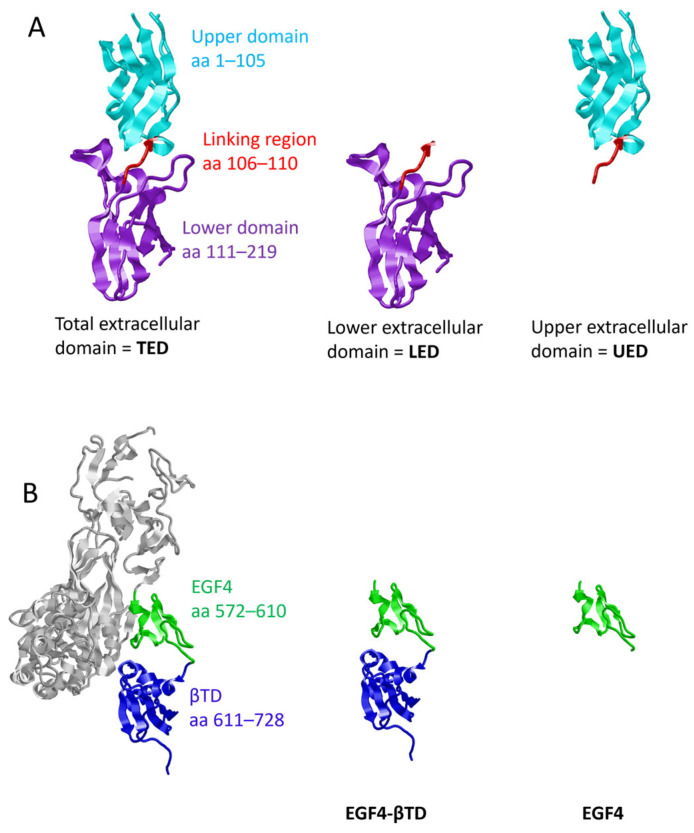
Structure of TF and β1-integrin protein constructs. (**A**) Depictions of the sections of the TF protein present in the TED (residues 1–219), LED (residues 106–219) and UED (residues 1–110) protein constructs, produced using the crystal structure of the extracellular domain of TF (1TFH). (**B**) β3-integrin (4G1E) was used as a proxy for β1-integrin, as the crystal structure of β1-integrin has not been published. In β3-integrin, the EGF4 spans residues 563–603 and the βTD residues 604–695. The corresponding domains within β1-integrin were estimated by homology comparison, with the EGF4 domain estimated to span residues 572–610, and βTD was localised to residues 611–728.

**Figure 2 cancers-17-00644-f002:**
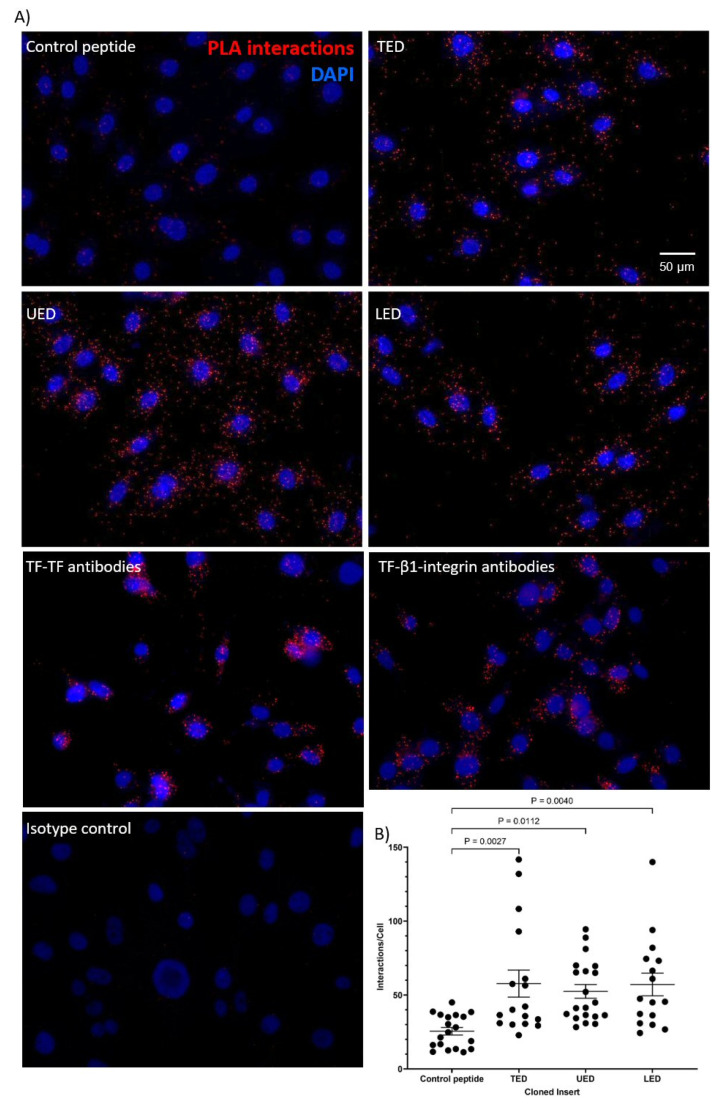
Association of TF extracellular domain peptide constructs with cellular β1-integrin in HDBEC. HDBEC (5 × 10^4^) were transfected to express TED, LED or UED constructs or control peptides. The interactions between the expressed peptides and β1-integrin were assessed by PLA using mouse anti-HA-tag (C29F4) and rabbit anti-β1-integrin antibodies. Positive controls were prepared using mouse anti-TF (HTF1) and rabbit anti-TF (FL295) antibodies, as well as mouse anti-TF (HTF1) and rabbit anti-β1-integrin antibodies. A negative control was prepared using the mouse anti-HA-tag antibody (C29F4) paired with a rabbit IgG isotype control antibody. (**A**) The cells were examined by fluorescence microscopy at ×40 magnification. (**B**) The number of red fluorescent interaction events and blue cell nuclei in each field of view was quantified using ImageJ.

**Figure 3 cancers-17-00644-f003:**
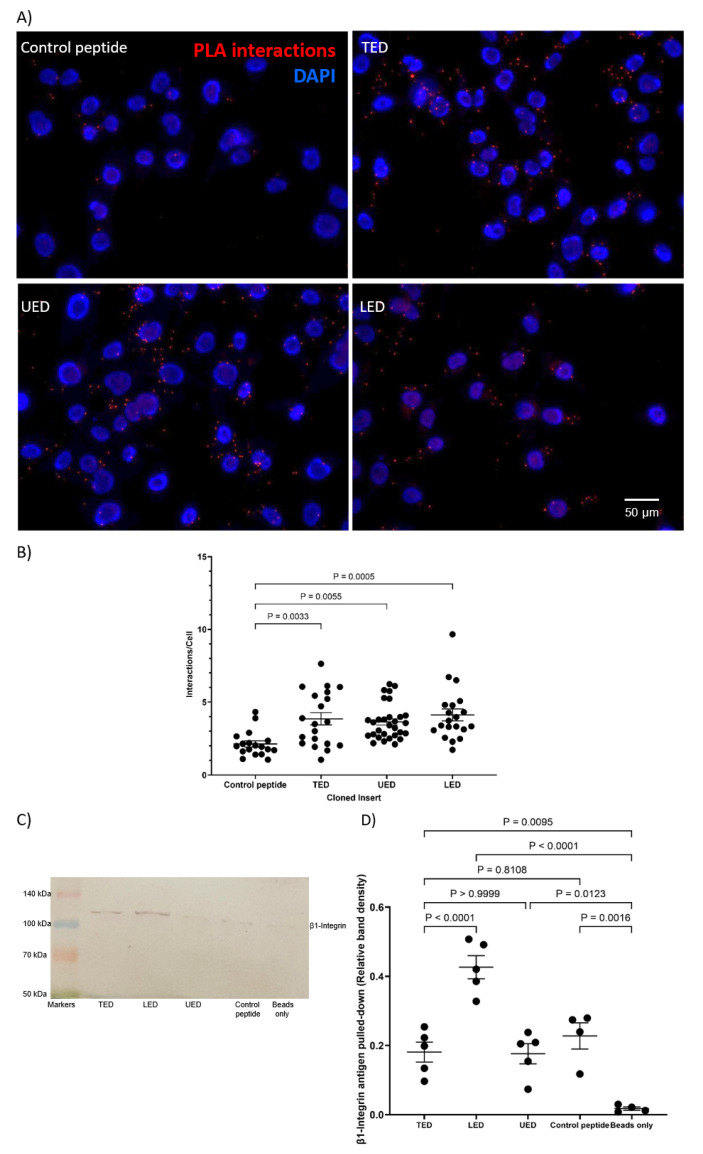
Association of TF-derived peptides with cellular β1-integrin in MDA-MB-231 cells. MDA-MB-231 cells (10^4^) were transfected to express the TED, LED or UED constructs or control peptide. The interactions between the expressed constructs and β1-integrin were assessed by PLA using mouse anti-HA-tag (C29F4) and polyclonal rabbit anti-β1-integrin antibodies. (**A**) The cells were examined by fluorescence microscopy at ×40 magnification. (**B**) The number of red fluorescent interaction events and blue cell nuclei in each field of view was quantified using ImageJ. MDA-MB-231 cells (1.5 × 10^5^) were transfected to express TED, LED or UED constructs or control peptide for 48 h and were then lysed. The recombinant protein constructs were immunoprecipitated from the lysates using the anti-FLAG-tag (4 µg; C29F4) antibody and protein A-magnetic beads. A sample of lysate was also incubated with the protein A-beads. (**C**) Western blot analysis of samples co-precipitated along with the TED, LED or UED constructs was carried out using a polyclonal anti-β1-integrin antibody. Images represent 3 separate experiments. (**D**) The amounts of β1-integrin precipitated from the samples were measured using ImageJ.

**Figure 4 cancers-17-00644-f004:**
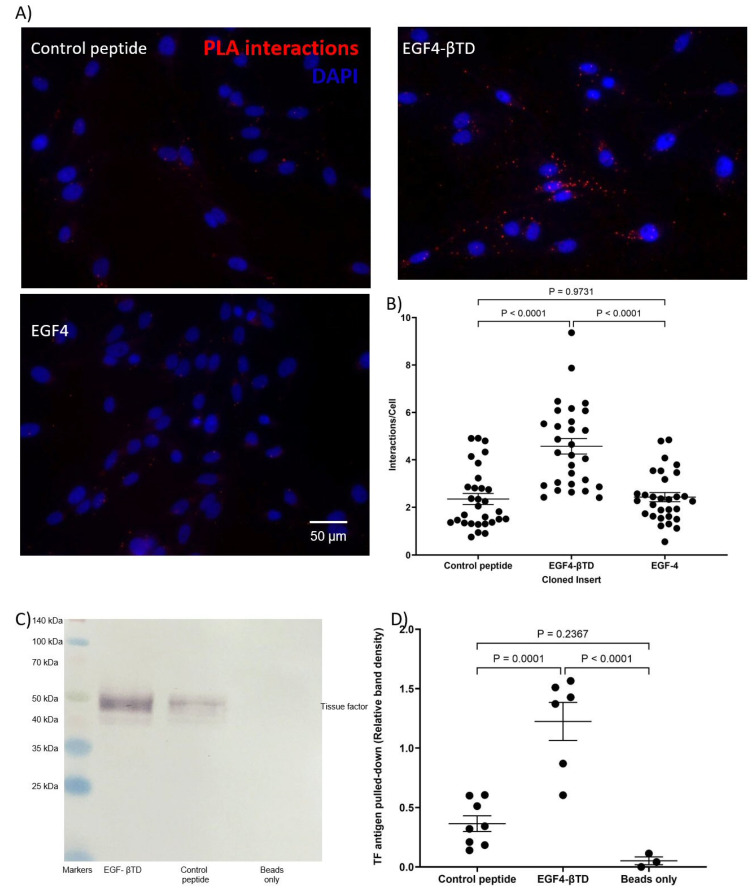
Association of EGF4-βTD peptide construct with cellular TF. MDA-MB-231 cells (10^4^) were transfected to express the EGF4-βTD or EGF4 constructs or control peptide. The interactions between the expressed constructs and TF were assessed by PLA using mouse anti-HA-tag (C29F4) and polyclonal rabbit anti-TF antibodies. (**A**) The cells were examined by fluorescence microscopy at ×40 magnification. (**B**) The number of red fluorescent interaction events and blue cell nuclei in each field of view was quantified using ImageJ. MDA-MB-231 cells (1.5 × 10^5^) were transfected to express the EGF4-βTD construct or control peptide for 48 h and were then lysed. The recombinant protein constructs were immunoprecipitated from the lysates using the anti-FLAG-tag (4 µg; C29F4) antibody and protein A-magnetic beads. A sample of lysate was also incubated with the protein A-beads. (**C**) Western blot analysis of samples precipitated along with the EGF4-βTD construct was carried out using an anti-TF antibody (HTF-1). Images represent 3 separate experiments. (**D**) The amounts of TF precipitated from the samples were measured using ImageJ.

**Figure 5 cancers-17-00644-f005:**
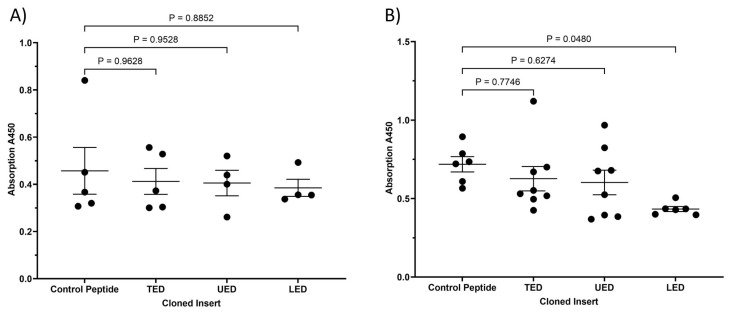
Analysis of the availability of the EGF4-βTD domain following expression of TF peptides. (**A**) MDA-MB-231 cells (10^4^) and (**B**) HDBEC (5 × 10^4^) were transfected to express TED, LED, UED or control peptides. The cells were probed with an antibody against residues 579–799 of β1-integrin (EGF4 and βTD domains). The cells were then probed with HRP-conjugated anti-rabbit IgG antibody and developed using TMB-one solution HRP substrate. The amount of bound antibody was quantified by measuring the absorptions at 450 nm using a plate reader. Values were normalised to the number of cells in each well, as determined by crystal violet assay measurements.

**Figure 6 cancers-17-00644-f006:**
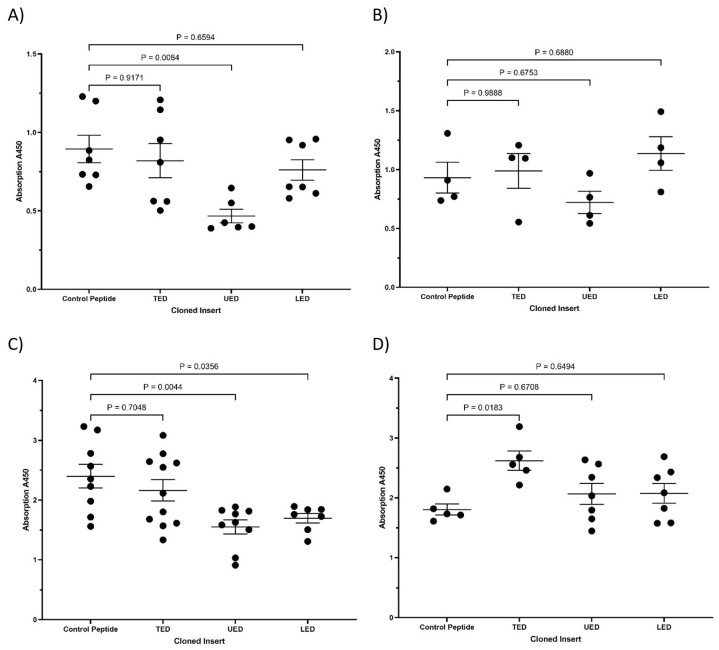
The influence of TF extracellular domain peptide constructs on the conformation of cellular β1-integrin. (**A**,**B**) MDA-MB-231 cells (10^4^) and (**C**,**D**) HDBEC (5 × 10^4^) were transfected to express TED, LED, UED or control peptides. The cells were probed with antibodies that specifically recognised the (**A**,**C**) active/open (9EG7) or (**B**,**D**) inactive/closed (AIIB2) conformation of β1-integrin. The cells were then probed with HRP-conjugated anti-rat IgG or anti-rabbit IgG antibodies and developed using TMB-one solution HRP substrate. The amount of bound antibody was quantified by measuring the absorptions at 450 nm using a plate reader. Values were normalised to the number of cells in each well, as determined by crystal violet assay measurements.

**Figure 7 cancers-17-00644-f007:**
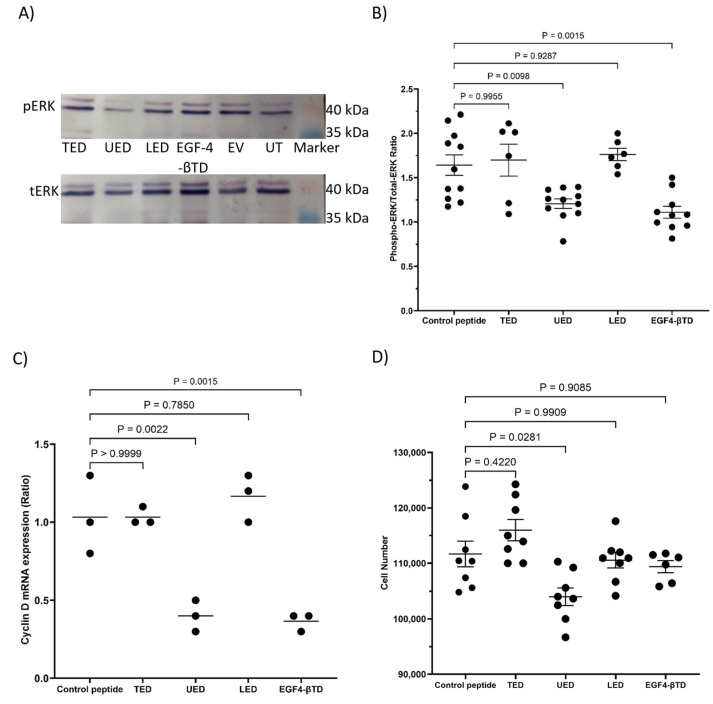
The influence of TF extracellular domain peptide constructs on proliferative signalling in MDA-MB-231 cells. MDA-MB-231 cells (1.5 × 10^5^) were transfected to express TED, LED, UED, EGF4-βTD or the control peptide. (**A**) Cells were lysed with Laemmli buffer, and Western blot analysis was carried out using antibodies against total ERK1/2, phosphorylated ERK1/2 and GAPDH. Images represent 3 separate experiments. (**B**) Band intensities were quantified using ImageJ, and the ratio of pERK/tERK was calculated. (**C**) Cells were lysed, and the mRNA was extracted and converted to cDNA using cell-2-cDNA kit. The expression of cyclin D1 and β-actin mRNA were measured using RT-PCR and the relative cyclin D1 expression levels calculated using the 2^−ΔΔCq^ method. (**D**) MDA-MB-231 cells (5 × 10^4^) were transfected to express TED, LED, UED, EGF4-βTD or control peptide. The cells were incubated for 72 h, and the number of cells was determined using the crystal violet assay.

**Figure 8 cancers-17-00644-f008:**
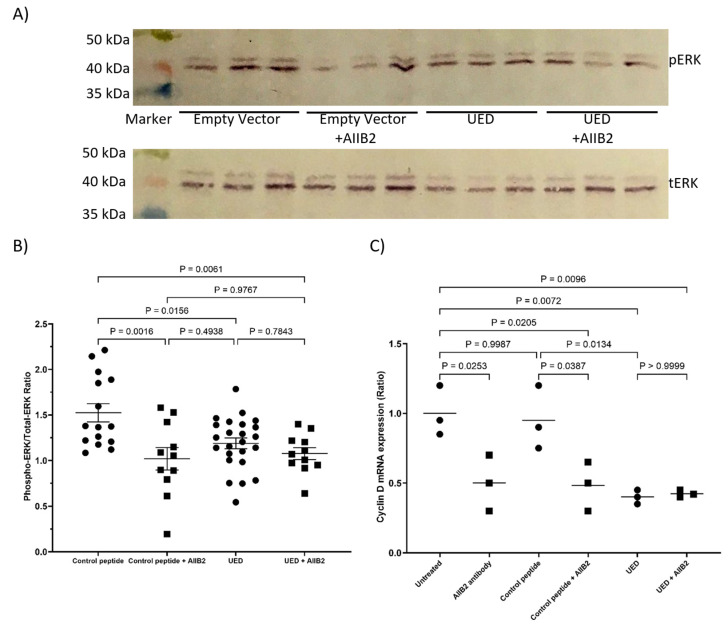
The influence of an inhibitory anti-β1-integrin antibody on the changes in proliferative signalling caused by TF extracellular domain peptide constructs. MDA-MB-231 cells (1.5 × 10^5^) were transfected to express UED or the control peptide. The cells were pre-incubated with an inhibitory anti-β1-integrin antibody (AIIB2; 10 μg/mL) for 16 h prior to collection. (**A**) Cells were lysed with Laemmli buffer and western blot analysis was carried out using antibodies against total ERK1/2, phosphorylated ERK1/2 and GAPDH. Images represent 4 separate experiments. (**B**) Band intensities were quantified using ImageJ and the ratio of pERK/tERK was calculated. (**C**) Cells were lysed, and the mRNA extracted and converted to cDNA using cell-2-cDNA kit. The expression of cyclin D1 and β-actin mRNA were measured using RT-PCR and the relative cyclin D1 expression levels calculated using the 2^−ΔΔCq^ method.

**Figure 9 cancers-17-00644-f009:**
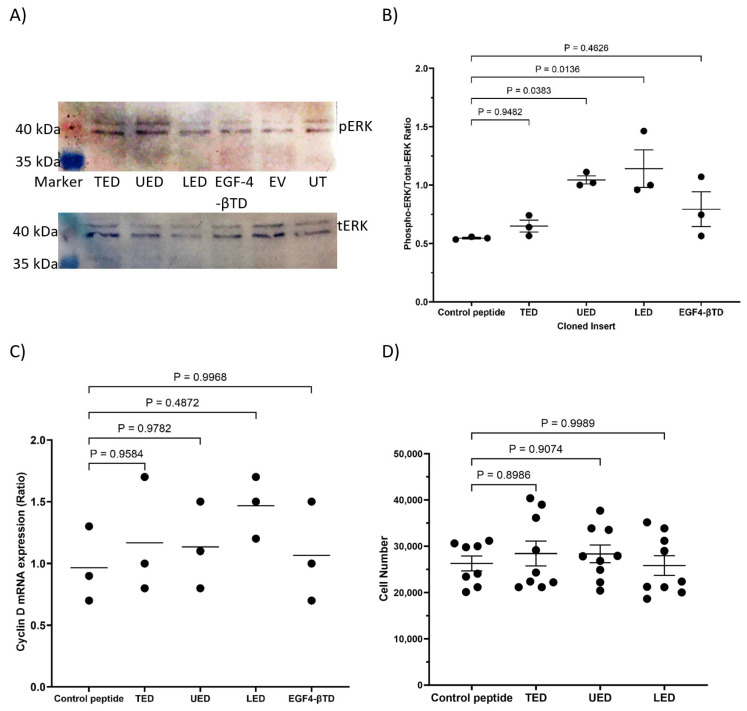
The influence of TF extracellular domain peptide constructs on proliferative signalling in HDBEC. HDBEC (5 × 10^5^) were transfected to express TED, LED, UED, EGF4-βTD or the control peptide. (**A**) Cells were lysed with Laemmli buffer, and Western blot analysis was carried out using antibodies against total ERK1/2, phosphorylated ERK1/2 and GAPDH. Images represent 2 separate experiments. (**B**) Band intensities were quantified using ImageJ and the ratio of pERK/tERK was calculated. (**C**) Cells were lysed, and the mRNA extracted and converted to cDNA using cell-2-cDNA kit. The expression of cyclin D1 and β-actin mRNA were measured using RT-PCR and the relative cyclin D1 expression levels calculated using the 2^−ΔΔCq^ method. (**D**) HDBEC (5 × 10^4^) were transfected to express TED, LED, UED, EGF4-βTD or control peptide. The cells were incubated for 72 h and the number of cells were determined using the crystal violet assay.

**Figure 10 cancers-17-00644-f010:**
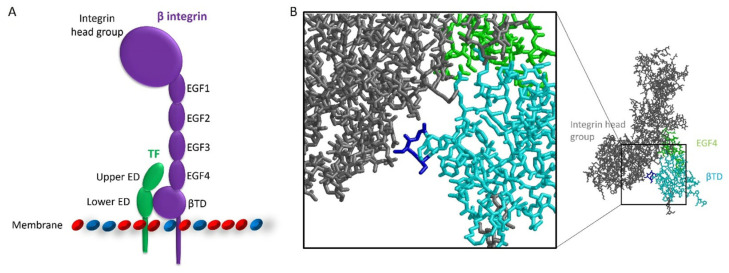
Proposed model for the interaction between TF and β1-integrin. (**A**) The proposed positioning for the interacting domains during complex formation between TF (green) and β1-integrin (purple). In the model, the upper region of TF extracellular domain interacts with the EGF4 domain of β1-integrin, and the lower region with the βTD domain. (**B**) The structural loop (blue) within the βTD (cyan) of a β3-integrin interacts with residues in the head group of the integrin and may hold the integrin in a closed configuration.

**Table 1 cancers-17-00644-t001:** Comparison of the influence of protein constructs in TF^+^MDA-MB-231 and TF^−^HDBEC.

	MDA-MB-231 (TF^+^)	HDBEC (TF^−^)
PLA	TED, LED, UED interact with β1-integrin. EGF4-βTD interacts with TF.	TED, LED, UED interact with β1-integrin.
IP	LED IP with β1-integrin. EGF4-βTD IP with TF.	N/A
ERK phosphorylation	UED and EGF4-βTD reduce ERK phosphorylation.	UED and LED increase ERK phosphorylation.
Cyclin D1 expression	UED and EGF4-βTD reduce cyclin D expression.	No significant changes.
Proliferation	UED reduces cell proliferation.	No significant changes.

## Data Availability

Data are contained within the article and Appendix A.

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
