# Peer review of "Identification of the Interacting Domains Between Tissue Factor and β1-Integrin and the Signalling Properties of the Two Fibronectin-like Domains of Tissue Factor"

_cancers, 2025, doi:10.3390/cancers17040644_

Round 1

Reviewer 1 Report

Comments and Suggestions for Authors

The authors investigated the interacting domains between TF and beta1-integrin and the major downstream effects of their interactions. The significance of the interaction between TF and integrin is well known. However, the details of the interaction are very complex and cannot be achievable by one experiment. The author's work is alinged with this continuous effort to identify the details of the interactions. Although the suggestions the authors made should be confirmed by further studies, I think their suggestions are highly probable.

Author Response

The authors investigated the interacting domains between TF and beta1-integrin and the major downstream effects of their interactions. The significance of the interaction between TF and integrin is well known. However, the details of the interaction are very complex and cannot be achievable by one experiment. The author's work is alinged with this continuous effort to identify the details of the interactions. Although the suggestions the authors made should be confirmed by further studies, I think their suggestions are highly probable.

We thank the reviewer for taking the time to review our manuscript and for the encouraging remarks. We fully agree that the interaction of TF and β1-integrin is extremely complex and involves multiple signalling pathways, of which we have only looked at one. Our aim in this study was to explore the protein-protein interaction mechanisms which contribute to one aspect (proliferation) of this interaction. We have now included additional sentences in the discussion (starting on line 522) to emphasise the contribution of our work, the specificity of the outcomes measured and to highlight the complexity of the interaction of TF with β1-integrin.

Reviewer 2 Report

Comments and Suggestions for Authors

The authors have shown the domains responsible for the interaction between TF and β1-integrin leading to increased cancer cell proliferation. Further, the authors have identified domain interactions that strengthen the TF-integrin complex formation. These mechanistic results suggest how and why tumors with high TF levels exhibit aggressive phenotypes. 

· Since the study focuses on the role of TF-β1 integrin in cancer, the article's title should accurately describe the diseases. Please make the appropriate edits.

 · Line 49 requires a reference.

 · Line 254 – To start with, the expression and lack of TF in MB231 cells and in HDBEC cells, at the protein level, need to be shown through western blots. Likewise, β1-integrin expression must be demonstrated in both cell types.

 · Lines 258–259: The protein is not necessarily glycosylated even if the observed molecular weight is greater than the calculated/original one. Indeed, glycosylation is a frequent alteration; however, a higher observed molecular weight than the original one can also result from other post-translational modifications such as phosphorylation or protein aggregation. The statement needs to be edited. It will be beneficial to do western blots for the same to make such conclusion.

 · Similar to Fig 2C, wherein all three constructs of TED, LED, or UED were used to perform the co-IP, it will be valuable to show the co-IP for EGF4 with β1-integrin, along with EGF-βTD. This will further validate the results from the PLA and confirm that it is the EGF4-βTD and not the EGF4 domain alone that is interacting with TF.

 · For all ERK phosphorylation data, every blot must be shown in the results section. Since band intensity measurement is a quantified representation of western blots, it cannot be used solely for interpretation. Blot images are absolutely necessary for the western blot technique.

 · Sentence ending in Line 399 needs a reference.

 · Why MB231 cells? Please explain the reasoning behind the cell selection. I think it is almost necessary to repeat some of the main experiments with cells from different cancer types, like pancreatic or cervical cancers, which have the highest expression of TF among solid tumors. This will strengthen the hypothesis.

 · At the end of each result section, in a few sentences, please explain the key takeaway from that particular set of experiments. For example: “Taken together, these results suggest that….”

 · Result 3.3 – Why there is a reduced ERK phosphorylation, cyclin D1 expression, and proliferation. Although the concept of competition has been introduced in the Discussion section, it needs to be mentioned here first, explaining why there is a reduction in phosphorylation and proliferation if these are the domains that are responsible for the interaction and thus increased proliferation.

 · At the end of the discussion, the authors must highlight the significance of their findings. In what ways will the field benefit from understanding the interaction domains between integrin and TF? Can they be next targeted to stop the interaction? What is the future from here? (in a few sentences). In other words, why knowing the interacting domains is important to targeting cancer cells.

Author Response

Reviewer 3

The authors have shown the domains responsible for the interaction between TF and β1-integrin leading to increased cancer cell proliferation. Further, the authors have identified domain interactions that strengthen the TF-integrin complex formation. These mechanistic results suggest how and why tumors with high TF levels exhibit aggressive phenotypes. 

We thank the reviewer for their time and patience in reviewing our manuscript. We have addressed all of their comments and hope that we have responded to the reviewer adequately.

  • Since the study focuses on the role of TF-β1 integrin in cancer, the article's title should accurately describe the diseases. Please make the appropriate edits.

The purpose of our study was to explore the outcome and contribution of protein-protein interaction on the proliferative signalling in the cells. Although this is an important factor in cancer growth, it represents only one aspect of the interaction, which is likely to play a role in other physiological and pathological contexts as well. We have carried out the work in two contrasting cell types with very high, and no TF, in order to be able to draw objective conclusions regarding proliferative signalling. As clearly pointed out by reviewer 1 above, the “details of the interaction are very complex and cannot be achievable by one experiment”. We therefore feel that the title of the manuscript is factual and deliberately limited to what we can clearly demonstrate. Additionally, in agreement with reviewer 1, since the interaction leads to many complex outcomes (invasion, metastasis, chemoresistance, etc), any reference to specific diseases would be very speculative at this stage. We therefore feel that any reference of specific diseases, would be deliberately misleading at this stage. However, we feel that the objectives and our findings are complementary to objectives of the special issue “Factors Regulating Cancer Cell Growth, …”. We would like to maintain the title as it is, since we do not feel comfortable making any more specific claims that we cannot justify at present.

  • Line 49 requires a reference.

A reference has been added.

  • Line 254 – To start with, the expression and lack of TF in MB231 cells and in HDBEC cells, at the protein level, need to be shown through western blots. Likewise, β1-integrin expression must be demonstrated in both cell types.

A new supplimentary figure (Figure S1) has been prepared showing the expression levels of both TF and β1-integrin in MDA-MB-231 cells and HDBEC.

  • Lines 258–259: The protein is not necessarily glycosylated even if the observed molecular weight is greater than the calculated/original one. Indeed, glycosylation is a frequent alteration; however, a higher observed molecular weight than the original one can also result from other post-translational modifications such as phosphorylation or protein aggregation. The statement needs to be edited. It will be beneficial to do western blots for the same to make such conclusion.

Thank you for this observation. We hypothesised that the high molecular weights were due to glycosylation because glycosylation is one of the most common post-translational modifications for extracellular domains of transmembrane proteins, whereas other modification such as phosphorylation are more common in intracellular proteins/domains of proteins. However, we agree we had no definitive evidence that the protein constructs are glycosylated. Therefore we have changed the sentence from “The observed molecular weights of the protein constructs . . . were glycosylated“ to “of the protein constructs . . . had undergone post-translational modification.” in line 258-260 of the revised manuscript.

  • Similar to Fig 2C, wherein all three constructs of TED, LED, or UED were used to perform the co-IP, it will be valuable to show the co-IP for EGF4 with β1-integrin, along with EGF-βTD. This will further validate the results from the PLA and confirm that it is the EGF4-βTD and not the EGF4 domain alone that is interacting with TF.

A new panel has been added to supplimentary figure 3 (Figure S3C). The figure shows the Co-IP of β1-integrin with LED and compares this to the lack of any detectable β1-integrin when using EGF-4, as is requested by the reviewer. Additionally, MDA-MB-231 cell lysate, which was used as the input material for the co-IP experiments, was also included in the western blot as requested by another reviewer.

  • For all ERK phosphorylation data, every blot must be shown in the results section. Since band intensity measurement is a quantified representation of western blots, it cannot be used solely for interpretation. Blot images are absolutely necessary for the western blot technique.

Images of the western blot membranes probe for total and phosphorylated ERK have now been added to Figure 6 and Figure 7 (Figures 7-9 in the updated manuscript). This required Figure 6 to be separated into two figures in order for all labels and axes to remain legible.  

  • Sentence ending in Line 399 needs a reference.

A reference has been added.

  • Why MB231 cells? Please explain the reasoning behind the cell selection. I think it is almost necessary to repeat some of the main experiments with cells from different cancer types, like pancreatic or cervical cancers, which have the highest expression of TF among solid tumors. This will strengthen the hypothesis.

Although breast tumours on average do not express as much TF as pancreatic tumours, absolute expression of TF in individual cancer cell lines depends less on the tissue of origin and more on the grade and aggressiveness of the cells. Prior to the study, we assessed the expression of TF in a range of both breast and pancreatic cell lines (see uploaded file). MDA-MB-231 cells, being a particularly aggressive triple-negative breast cancer cell line, expressed similar levels of TF to the pancreatic cell line BxPC-3 and more than Panc-1 and AsPC-1 cells. Similarly, in the article Ettelaie, C. et al. Thrombosis J 14, 2 (2016) doi.org/10.1186/s12959-016-0075-3, cell lines from an even wider range of tissues (including breast, pancreas, melanoma, colorectal and lung) were assessed for TF expression and activity. Again, MDA-MB-231 cells were found to possess some of the highest TF mRNA expression levels and the highest TF protein expression among the 17 cell lines tested.

  • At the end of each result section, in a few sentences, please explain the key takeaway from that particular set of experiments. For example: “Taken together, these results suggest that….”

Concluding sentences have been added to all 3 results paragraphs. These start on lines 312, 328 and 380 of the revised manuscript.

  • Result 3.3 – Why there is a reduced ERK phosphorylation, cyclin D1 expression, and proliferation. Although the concept of competition has been introduced in the Discussion section, it needs to be mentioned here first, explaining why there is a reduction in phosphorylation and proliferation if these are the domains that are responsible for the interaction and thus increased proliferation.

The concept of competition between the protein constructs and cellular TF and β1-integrin has now been introduced in the results section. The sentence “The reduction in proliferative signalling may result from the UED and EGF4-βTD constructs competing with the binding of the exogenously expressed TF and β1-integrin in the MDA-MB-231 cells” has been added starting on line 365 of the revised manuscript.

  • At the end of the discussion, the authors must highlight the significance of their findings. In what ways will the field benefit from understanding the interaction domains between integrin and TF? Can they be next targeted to stop the interaction? What is the future from here? (in a few sentences). In other words, why knowing the interacting domains is important to targeting cancer cells.

As stated above, we have been prudent to avoid any unjustified claims. The interactions of TF with different integrins, and other proteins have profound influence on cancer cells and although established, is still debated in the literature. Our study has demonstrated the putative function for each of the two domains and has identified the potential of domains as mediators in inducing proliferative signalling. Since the recombinant domains were able to reduce the proliferative signal in MDA-MB-231 cells, it may be envisaged that the expression of the recombinant proteins may have a similar impact in vivo. However, it is more likely that identification of the amino acids involved in the interaction between TF and β1-integrin would permit a more targeted inhibition of proliferative property associated with TF. For example, targeted antibodies, or small disrupting molecules may be used to reduce cell proliferation in diseases such as cancer or alternatively to enhance growth in order to promote tissue repair. We have now included some of the above in a new paragraph at the end of the discussion section (starting on line 522).   

Reviewer 3 Report

Comments and Suggestions for Authors

This manuscript aims to identify the interacting domains between tissue factor (TF) and b1-integrin. They use deletion mutants in PLA and IP assays to confirm the upper fibronectin-like domain of TF interacts with the EGF4 domain of b1-integrin. The interaction may alter cell proliferation by ERK activity and cyclin D analysis. I have some minor comments for the data presentation to make it easy to read.

(1)   Make a figure to show the structure domains of TF and b1-integrin and indicate the deletion constructs.

(2)   All quantitative plot labels were too small to read.

(3)   The IP results of Fig.2C and 3C should include input control; Fig.2D and 3D ImageJ quantitative results should describe how many experiments were analyzed in legend.

(4)I suggest using a table summarizing the PLA, IP, kinase, and cyclin D expression results in MDA-MB-231 and HBDEC cells.

Author Response

This manuscript aims to identify the interacting domains between tissue factor (TF) and b1-integrin. They use deletion mutants in PLA and IP assays to confirm the upper fibronectin-like domain of TF interacts with the EGF4 domain of b1-integrin. The interaction may alter cell proliferation by ERK activity and cyclin D analysis. I have some minor comments for the data presentation to make it easy to read.

We thank the reviewer for taking the time to review our manuscript and for their suggested improvements. We have detailed how we have implicated each of their suggestions below.

(1) Make a figure to show the structure domains of TF and b1-integrin and indicate the deletion constructs.

A new figure (Figure 1) depicting the domains present in each of the protein constructs has now been added to the manuscript.

(2) All quantitative plot labels were too small to read.

All figures have now been updated to enlarge the font of all the figure labels and axes, hopefully this will improve the legibility of the figures.

(3) The IP results of Fig.2C and 3C should include input control; Fig.2D and 3D ImageJ quantitative results should describe how many experiments were analysed in legend.

The number of experiments analysed has now been added to the legends of all figures containing representative images of western blots (Figures 2-3 and 7-9). Additionally, we have now demonstrated the presence of β1-integrin in cell lysate from MDA-MB-231 cells in the supplemental figure 3C. The cell lysate used was the input material for the co-IP experiments.

(4) I suggest using a table summarizing the PLA, IP, kinase, and cyclin D expression results in MDA-MB-231 and HBDEC cells.

A table summarizing the PLA, IP, kinase, and cyclin D expression results has been added to page 11 of the manuscript.

Round 2

Reviewer 2 Report

Comments and Suggestions for Authors

The authors have successfully addressed my concern and suggestions.